# Vitamins D and E Stimulate the *PI3K-AKT* Signalling Pathway in Insulin-Resistant SK-N-SH Neuronal Cells

**DOI:** 10.3390/nu11102525

**Published:** 2019-10-19

**Authors:** Amirah Salwani Zaulkffali, Nurliyana Najwa Md Razip, Sharifah Sakinah Syed Alwi, Afifah Abd Jalil, Mohd Sokhini Abd Mutalib, Banulata Gopalsamy, Sui Kiat Chang, Zaida Zainal, Nafissa Nadia Ibrahim, Zainul Amiruddin Zakaria, Huzwah Khaza’ai

**Affiliations:** 1Department of Nutrition and Dietetics, Faculty of Medicine and Health Sciences, Universiti Putra Malaysia, Serdang 43400, Malaysia; amirahsalwani0291@gmail.com (A.S.Z.); sokhini@medic.upm.edu.my (M.S.A.M.); 2Department of Biomedical Sciences, Faculty of Medicine and Health Sciences, Universiti Putra Malaysia, Serdang 43400, Malaysia; najwabiochem@gmail.com (N.N.M.R.); sh_sakinah@upm.edu.my (S.S.S.A.); afifahjalil91@yahoo.com (A.A.J.); banulatagopalsamy@gmail.com (B.G.); nafissanadia@yahoo.com.my (N.N.I.); zaz@upm.edu.my (Z.A.Z.); 3Department of Nutrition and Dietetics, School of Health Sciences, International Medical University, Kuala Lumpur 57000, Malaysia; suikiatchang@gmail.com; 4Nutrition Unit, Product Development and Advisory Services Division, Malaysian Palm Oil Board, Bandar Baru Bangi 43000, Malaysia; zaida@mpob.gov.my

**Keywords:** insulin resistance, vitamin D, vitamin E, SK-N-SH neuronal cells, glucose uptake

## Abstract

This study investigated the effects of vitamins D and E on an insulin-resistant model and hypothesized that this treatment would reverse the effects of Alzheimer’s disease (AD) and improves insulin signalling. An insulin-resistant model was induced in SK-N-SH neuronal cells with a treatment of 250 nM insulin and re-challenged with 100 nM at two different incubation time (16 h and 24 h). The effects of vitamin D (10 and 20 ng/mL), vitamin E in the form of tocotrienol-rich fraction (TRF) (200 ng/mL) and the combination of vitamins D and E on insulin signalling markers (*IR*, *PI3K*, *GLUT3*, *GLUT4,* and *p-AKT)*, glucose uptake and AD markers (*GSK3β* and *TAU*) were determined using quantitative real-time polymerase chain reaction (qRT-PCR) and enzyme-linked immunosorbent assay (ELISA). The results demonstrated an improvement of the insulin signalling pathway upon treatment with vitamin D alone, with significant increases in *IR*, *PI3K*, *GLUT3*, *GLUT4* expression levels, as well as *AKT* phosphorylation and glucose uptake, while *GSK3β* and *TAU* expression levels was decreased significantly. On the contrary, vitamin E alone, increased *p*-*AKT*, reduced the ROS as well as *GSK3β* and *TAU* but had no effect on the insulin signalling expression levels. The combination of vitamins D and E only showed significant increase in *GLUT4*, *p-AKT*, reduced ROS as well as *GSK3β* and *TAU.* Thus, the universal role of vitamin D, E alone and in combinations could be the potential nutritional agents in restoring the sensitivity of neuronal cells towards insulin and delaying the pathophysiological progression of AD.

## 1. Introduction

Insulin resistance and neuronal dysfunction are strongly correlated with Alzheimer’s disease (AD), as demonstrated by the Rotterdam study [1]. A dramatic increase in the worldwide incidence of diabetes is directly proportional to age. Moreover, age is a risk factor associated with diabetes and neurodegenerative diseases such as AD [2]. Post-mortem analysis of brains revealed that the molecular, biochemical, and signal transduction abnormalities in AD are identical to those that occur in type 1 diabetes mellitus (T1DM) and type 2 diabetes mellitus (T2DM) [2,3]. Decreased cerebrospinal fluid (CSF) insulin levels and reduced insulin-mediated glucose disposal were observed in AD patients compared to those observed in healthy subjects [4]. A cascade of reactions initiated by the reduced sensitivity of cells to insulin resulted in the reduced insulin level in the brain of AD patients. Reduced sensitivity to insulin can result in hyperinsulinaemia, which consequently downregulates the insulin receptors at the blood-brain barrier, impairing insulin signalling and reducing the uptake of insulin by brain cells. These events later manifest as brain inflammation, oxidative stress, alterations in beta-amyloid (Aβ) levels, and cell death [5]. Insulin resistance in brain is also named as ‘type 3 diabetes’ [6]. Since insulin resistance can be a component of AD pathogenesis, it could be a potent accelerator of AD. There are numerous aetiologies for insulin resistance, including lipotoxicity, inflammation, oxidative stress and hyperinsulinaemia [7]. Hyperinsulinaemia contributes to insulin resistance, and its effects on peripheral tissues, such as skeletal muscle, liver, and adipose tissue, have been demonstrated [8]. The involvement of neurons in brain insulin resistance is relatively unclear, since little is known about its molecular mechanism.

The role of vitamins D and E in the pathogenesis and prevention of diabetes has generated enormous scientific interest. The association between vitamin D status and the risk of diabetes or glucose intolerance has been known since the early 1990s [9]. It has been shown that the prevalence of hypovitaminosis D is higher in diabetic patients than in non-diabetic people [10,11,12]. Indeed, vitamin D improves insulin resistance and T2DM by affecting insulin sensitivity, β-cell function or both [13,14]. Vitamin D may modulate insulin sensitivity because a functional vitamin D receptor (VDR) element has been identified in the promoter region of the insulin receptor, suggesting that vitamin D regulates insulin transcription through genomic interactions [15]. Vitamin E is a fat-soluble compound with antioxidant properties that exist in eight forms in nature (alpha-, beta-, gamma- and delta-tocopherol and alpha-, beta-, gamma- and delta- tocotrienol) each with its own biological properties [16]. The difference between tocopherol and tocotrienol is that they have different number and position of methyl groups attached to the aromatic ring [17]. Vitamin E exhibits antioxidant properties that improve the insulin signalling cascade in both *in vitro* and *in vivo* models [18,19,20]. Tocotrienol occurs at very low levels in nature, with the highest concentration found in palm oil. Currently, there is an increase of interests on tocotrienol rich fraction (TRF) from palm oil. TRF consist of 25% of alpha-tocopherol (α-TCP) and 75% of tocotrienol [16]. Hence, vitamin E in the form of tocotrienol-rich fraction (TRF) may also boost insulin sensitivity and decrease diabetes risk by quenching free radicals and simultaneously reducing oxidative stress in the body [21]. Although there are many factors that lead to the development of AD, this study focuses on insulin resistance as the causal factor that mimics AD in neuronal cells. It is anticipated that the results from this study would be useful to identify suitable remedies that help to reverse the condition of insulin resistance in AD. Therefore, this study aims to evaluate the potency of vitamins D and E in improving insulin resistance in neuronal-insulin resistance model. The potency of vitamins D and E in modulating insulin signalling cascade were assessed at the gene expression level. This study evaluates the gene expression of insulin signalling markers involved such as insulin receptor (*IR*), phosphatidylinositol-4,5-bisphosphate 3-kinase (*PI3K*), protein kinase B or (*AKT*), glucose transporter 3 (*GLUT3*) and glucose transporter 4 (*GLUT4*); Alzheimer’s disease (AD) markers such as glycogen synthase kinase-3 beta (*GSK3β*) and tau (*TAU*). Besides, glucose uptake was also carried out to substantiate the restoration of insulin signalling upon vitamins D and E supplementation. Finally, this study also determined the oxidative stress levels in the cells upon treatment with vitamins D and E under the condition of insulin resistance.

## 2. Materials and Methods

### 2.1. Materials

The neuroblastoma cell line, SK-N-SH (ATCC^®^ HTB-11^TM^), was purchased from American Type Cell Culture (ATCC) (Manassas, VA, USA). Cell culture flasks, serological pipettes, and microplates were obtained from Techno Plastic Products (Trasadingen, Switzerland). Minimum essential medium (MEM) culture media, penicillin/streptomycin, 0.25% trypsin, foetal bovine serum (FBS) and phosphate buffer saline (PBS) were purchased from Nacalai Tesque (Kyoto, Japan). *IR*, *PI3K*, *GLUT3*, *GLUT4*, *GSK3β*, *TAU* and glyceraldehyde-3-phosphate dehydrogenase (*GAPDH*) primers were purchased from Integrated DNA Technologies (Coralville, IA, USA). TBE buffer, TAE buffer and agarose powder were purchased from Amresco (Solon, OH, USA). Insulin and human recombinant *E. coli* were purchased from BioVision (San Francisco, CA, USA). 3-(4,5-dimethylthiazol-2-yl)2,5-diphenyltetrazolium bromide (MTT) powder was purchased from PhytoTechnology Laboratories (Flint St, KS, USA). Dimethyl sulfoxide (DMSO) and vitamin D in the form of vitamin 1,25(OH)_2_D_3_ were purchased from Sigma Aldrich (St. Louis, MO, USA). Vitamin E in the form of tocotrienol-rich fraction (TRF) was supplied by Gold Tri E70 TRF, Sime Darby Research (Kuala Lumpur, Malaysia). The TRF content (25% α-tocopherol and 75% tocotrienols) and its purity were confirmed by our previous studies [22,23]. 

### 2.2. MTT Assay

Prior to developing an insulin resistance condition, an MTT assay was conducted to measure whether the induction with insulin induces toxicity to the cells. The function of the MTT assay is to cleave tetrazole rings in the functional mitochondria viable cells, producing insoluble dark purple formazan products. As a result, viable cells can be distinguished from dead cells. Ninety-six-well plates with a cell density of 2 × 10^5^ cells/mL were seeded for treatment with different concentrations of insulin. Cells were harvested in trypsin-EDTA after reaching a confluence of 70–80%. After an overnight incubation to allow cell attachment, insulin was added to the culture medium at the previously prepared concentrations of 100, 150, 200 and 250 nM for 16 and 24 h. After 30 min, the cells were re-challenged with 100 nM insulin for 30 min. A control without treatment (0 nM insulin) was also included. The previous media was removed, and the wells were washed three times with 200 µL PBS. Then, 200 µL treatment solutions were pipetted into the respective wells, and 200 µL serum-free media was used as blank. Twenty microliters of MTT solution was added to each well without removing the treatment solution. The plate was thoroughly shaken to evenly mix the contents. The plate was then covered with aluminium foil to avoid light penetration and incubated for 3 to 4 h before adding DMSO. After 3 h of incubation, all solutions in the wells that contained cells were completely removed by pipetting. Then, 100 μL DMSO solution was added to each well, including the blank wells. The quantity of formazan was measured by recording the change in absorbance at 570 nm using a microplate reader (Biochrom Asys UVM 340) with a reference wavelength of 630 nm and 15 s shaking time [24].

### 2.3. Cell Culture and Induction of Insulin Resistance

SK-N-SH is a neuroblastoma cell line that displays epithelial morphology and grows in adherent culture. This neuronal cell line was used to model an insulin resistance condition since it expresses the components of the insulin signalling pathway [25,26,27,28,29]. Cells at 70–80% confluence were harvested with trypsin-EDTA and 2 × 10^5^ cells were seeded in 96-well plates. After an overnight incubation, insulin was added to the culture medium beginning with the higher concentrations and progressing to the lower concentrations (100, 150, 200 and 250 nM) for 16 and 24 h. Cells were cultured in MEM supplemented with 10% FBS and antibiotics (penicillin 100 IU/ml, streptomycin 100 µg/ml) and were incubated at 37 °C and 5% CO_2_. Cells within three passages that were stored in liquid nitrogen were used for further experiments. The insulin resistance model was developed by different insulin concentrations in the range of 100–250 nM, as described in previous studies [28,30] with slight modifications. Briefly, serum-starved cells were incubated for 16 and 24 h at 37 °C in serum-free MEM in the absence or presence of insulin. After induction, the cells were re-challenged with 100 nM insulin in serum-free media for 30 min to determine the responsiveness of the insulin receptor.

### 2.4. Preparation and Treatment of Vitamins D and E in the Insulin Resistance Model

Vitamin D_3_ (1,25(OH)_2_D_3_) and vitamin E were prepared in a laminar flow hood to maintain sterility. Both vitamin D and vitamin E (in the form of TRF) were weighed, dissolved in 100% ethanol and dried under nitrogen gas before being diluted to the desired concentrations. After developing the insulin resistance model, the cells were supplemented with vitamin D at doses of 10 ng/mL and 20 ng/mL, vitamin E in the form of TRF at a dose of 200 ng/mL and the combination of vitamin D (10 or 20 ng/mL) and E (200 µg/mL) for 24 h at 37 °C and 5% CO_2_. The concentrations of vitamins D and E were chosen according to the results of previous studies by [25,31], respectively. Our previous studies demonstrated that 200 ng/mL vitamin E in the form of TRF was the optimum concentration to produce treatment effects in neuronal cells [22,23].

### 2.5. Quantification of Insulin Signalling and AD Markers by qRT-PCR

The expression levels of insulin signalling markers, such as *IR*, *PI3K*, *GLUT3* and *GLUT4*, are closely related to the impairment of insulin signalling that is caused by prolonged exposure to insulin (hyperinsulin). In addition, the expression of the AD markers *GSK3β* and *TAU* were also examined to understand the link between insulin resistance and Alzheimer’s disease. The expression of all markers was quantified using real-time PCR (RT-PCR). Total RNA was isolated using a FavorPrep™ Blood/Cultured Cell Total RNA Mini Kit (Favorgen, Taiwan) according to the manufacturer’s instructions. cDNA synthesis was performed using a qPCRBIO cDNA Synthesis kit (PCR Biosystems, London, UK), following the manufacturer’s instructions. The isolated RNA was then reversed transcribed to cDNA by RT-PCR. RT-PCR was conducted using a qPCRBIO cDNA Synthesis kit purchased from PCR Biosystems (London, UK) according to the manufacturer’s instructions. The primers targeting the genes of interest in this procedure that were used in this experiment are shown in Table 1. The PCR conditions for these primers were 95 °C for 1 min and 40 cycles of 95 °C for 5 s and 60 °C for 20 s. The sequences of all primers targeting the genes of interest were synthesized by First Base (Kuala Lumpur, Malaysia), designed using the Ape software (http://biologylabs.utah.edu/jorgensen/wayned/ape/) and blasted using NCBI primer-BLAST. The housekeeping gene for all primers was glyceraldehyde-3-phosphate dehydrogenase (*GAPDH*); the qPCR data were normalized to the GADPH values. Three independent experiments were performed, and the quantitative data obtained were averaged based on quantification cycle (C_q_) values, which were used to calculate the fold expression ratio.

### 2.6. Phosphorylation of AKT Assay

The phosphorylation of *AKT* assay was performed as described in the manual of Human Phospho-AKT Cell-Based ELISA kit (R & D Systems, Minneapolis, MN, USA). A total of 2 × 10^5^ cells was seeded in 96-well plates. After an overnight incubation, insulin was added to the culture medium beginning with the lower concentrations and progressing to the higher concentrations (100, 150, 200 and 250 nM dissolved in serum-free media) for 16 and 24 h. A control without treatment (serum-free media without insulin) was also included. The treated SK-N-SH cells were fixed with 4% formaldehyde for 20 min at 37 °C and washed three times with 200 µL 1X wash buffer for 5 mins each with gentle shaking. Subsequently, 100 μL quenching buffer was added and incubated for 20 min. The cells were further incubated for 1 h with blocking buffer and aspirated. One hundred microliters of phospho-Tie-2 antibody mixture (1:100) was then added to each well and incubated for 16 h at 4 °C. After incubation, the wells were washed three times with 1X wash buffer followed by the addition of 100 µL HRP conjugated antibody mixture (1:100) to each well. Then, the mixtures were incubated for 2 h at room temperature. The secondary antibody mixtures were removed, and the cells were washed twice with 200 µL 1X wash buffer, followed by two washes with 200 µL 1X PBS. The contents in the wells were aspirated, and 75 µL fluorogenic substrate for HRP was added into each well and incubated for 1 h at 37 °C. The plates were protected from direct light. Finally, the plate was analysed twice using a fluorometric plate reader (FLUOstar^®^ Omega, BMG LABTECH) at 450 nm/600 nm and 360 nm/450 nm excitation/emission wavelengths.

### 2.7. Glucose Uptake Assay

Glucose uptake was determined to demonstrate whether prolonged insulin treatment affects glucose transport into the cell. The glucose uptake assay was performed according to the manufacturer’s instructions as stated in the 2-D-Glucose (2-DG) Uptake Assay Kit purchased from Abnova (Taipei, Taiwan). A total of 2 × 10^5^ cells were seeded in 96-well plates. After the successful development of cellular insulin resistance, the cells were then treated with vitamin D (10 and 20 ng/mL), vitamin E (200 ng/mL) or a combination of vitamins D and E for 24 h. After the incubation, the media from each well was removed, and the cells were washed twice with 100 µL 1X Krebs-Ringer-Phosphate-HEPES (KRPH) buffer. After washing, 90 µL glucose uptake buffer was added and incubated for 1 h at 37 °C in a 5% CO_2_ incubator. Subsequently, 10 µL 2-DG solution was added into each well and incubated for 4 h at 37 °C and 5% CO_2_. Afterwards, the solution was removed and gently washed three times with KRPH buffer to remove the remaining 2-DG from the solution. To lyse the cells, 25 µL acidic lysis buffer was added to each well, and the mixture was incubated at 37 °C for 20 min. Then, 25 µL neutralization buffer was added to each well to neutralize the cell lysate for 10 min. Finally, 50 µL 2-DG uptake mixture was added to each well and incubated at room temperature for 1 h. The results were immediately read at excitation/emission wavelengths of 570/610 nm using a fluorometric plate reader (FLUOstar^®^ Omega, Biel/Bienne, Switzerland, BMG LABTECH).

### 2.8. ROS Measurement Assay

A commercially available dichlorodihydrofluorescein diacetate (DCFH-DA) dye (OxiSelect™ Intracellular ROS Assay Kit, Cell Biolabs) was used to measure the production of ROS according to the manufacturer’s instructions with slight modifications. A total of 2 × 10^5^ cells were seeded in 96-well plates. After the successful development of cellular insulin resistance, the cells were then treated with vitamin D (10 and 20 ng/mL), vitamin E (200 ng/mL) or a combination of vitamins D and E for 24 h. The treated cells were washed gently twice with Dulbecco’s phosphate-buffered (DPBS) saline (Thermo Fisher, Waltham, MA, USA) before 200 μL DCFH-DA dye was added to the cells. Then the cells were incubated at 37 °C for 1 h washed twice with DPBS. Media (200 μL) was added to the cells, followed by 200 μL 2X cell lysis buffer; then, the lysates were incubated for 5 min. One hundred fifty microliters of lysate was then transferred to a black 96-well plate, and the fluorescence was measured using a fluorometric plate reader (FLUOstar^®^ Omega, BMG LABTECH) at an excitation/emission wavelength of 480 nm/530 nm. The results were expressed in relative fluorescence units (RFUs) by plotting a standard curve of DCF concentration against RFU. The fluorescence intensity is proportional to the ROS levels in the cytoplasm.

### 2.9. Statistical Analysis

All independent experiments were conducted in triplicate. Statistical analyses were performed with one-way ANOVA followed by Dunnett’s multiple comparison test, with comparison to the control, using GraphPad Prism 5 Software for Windows (GraphPad Software, Inc., San Diego, CA, USA). All data were expressed as the mean ± SEM of three independent experiments, and significant values are expressed as * *p* < 0.05, ** *p* < 0.01 and *** *p* < 0.001 compared to the negative control group and # *p* < 0.05, ## *p* < 0.01 and *### p* < 0.001 compared to the control group.

## 3. Results

### 3.1. Effects of Prolonged Insulin Induction on Cell Viability

The viability of SK-N-SH cells was assessed via MTT assay upon exposure to 100–250 nM insulin for either 16 or 24 h. Experimental insulin treatment concentrations were chosen to maximize the insulin induction levels while minimizing cytotoxicity. Figure 1a,b shows the insignificant increase in cell viability upon the induction of insulin (100–250 nM) for 16 and 24 h compared to the control. This result indicates that all insulin concentrations used in the current study were unable to induce cell death or cell toxicity in neuronal cells. Hence, the effects of prolonged exposure to insulin concentrations up to 250 nM were safe with no adverse effects on neuronal SK-N-SH cells.

### 3.2. Establishment of Insulin Resistance Model in SK-N-SH Cell Line

#### 3.2.1. Analysis of Expression of Insulin Signalling Markers

The expression of insulin signalling markers were determined to validate the insulin resistance model adopted in the current study. The most upstream gene *IR* was found to be significantly reduced at insulin concentrations of 200 and 250 nM upon incubation at 16 h. Further attenuation of *IR* was observed in cells exposed to 100–250 nM insulin for 24 h with fold ratio declined to 0.25 (Figure 2a). Collectively, all other markers in insulin signaling cascade; *IR*, *PI3K*, *GLUT3,* and *GLUT4* were reduced significantly upon exposure to series of insulin concentrations. However, prominent and consistent attenuation of *PI3K/AKT* pathways was only achieved upon prolonged exposure to insulin for 24 h (Figure 2b–d). On the contrary, the expression of *GSK3β* increased significantly (*p* < 0.01) when exposed to high insulin concentration at 16 h but not at 24 h, as *GSK3β* activity is negatively regulated by insulin (Figure 2e).

#### 3.2.2. Phosphorylation of AKT in Insulin-Resistant Cells

Figure 3 shows that the level of AKT phosphorylation decreased signficantly (*p* < 0.05 and *p* < 0.01) in 16 and 24 h at 100–250 nM. At 16 h exposure, the highest reduction is achieved at 100 nM by 0.3 fold ratio. Similar signficant reduction (*p* < 0.05 and *p* < 0.01) was observed at 24 h exposure with all concentrations of insulin. Highest reduction by 0.5 fold ratio was achieved at 150 nM.

#### 3.2.3. Glucose Uptake in Insulin-Resistant Cells

The 2-DG uptake was reduced at all levels of insulin concentration (Figure 4) as compared to control. At 16 h exposure, 2-DG uptake was significantly decreased (*p* < 0.05) by 20% at 150–250 nM compared to control. At 24 h, the uptake of 2-DG decreased significantly (*p* < 0.05) by 20% at 100–250 nM. No marked changes between 16 and 24 h were observed, suggesting that exposure time has less influence on stimulating glucose uptake.

### 3.3. Expression of Insulin Signalling Markers upon Treatment with Vitamins D and E

Figure 5a–d show the relative transcriptional profiles of the key genes involved in insulin signalling upon intervention with vitamins D and E in insulin-resistant neuroblastoma SK-N-SH cell lines. High concentrations of insulin (250 nM) induced chronic insulin resistance in cells. This insulin resistance led to significant deterioration of insulin signalling markers *IR*, *PI3K*, *GLUT3* and *GLUT4* (*p* < 0.01). Supplementation with vitamin D significantly increased the expression of *IR*, *PI3K,* and *GLUT3* (*p* < 0.05) by 2 fold ratio compared to the negative control (Figure 5a–c). On the other hand, the combination of both vitamins did not improve the expression of *IR* and *PI3K* as well as treatment with vitamin E alone (Figure 5a,b). The expression levels of both *GLUT3* and *GLUT4* increased significantly (*p* < 0.05) to ~2 and ~1 fold ratio, respectively after treatment with vitamin D and the combination of both vitamins D and E (Figure 5c,d). Treatment with vitamin E alone did not increase the expression levels of *GLUT3* and *GLUT4* (Figure 5c,d).

### 3.4. Phosphorylation of AKT

Vitamin D (10 and 20 ng/mL), vitamin E and the combination of vitamins D and E increased *AKT* phosphorylation level significantly (*p* < 0.05) compared to the negative control (Figure 6) after 16 h induction of chronic insulin resistance. The *AKT* phosphorylation level was increased by 1-fold after treatment with vitamin D, vitamin E and a combination of both vitamins (Figure 6).

### 3.5. Glucose Uptake Assay

The induction of chronic insulin resistance impaired the uptake of 2-DG significantly (*p* < 0.001) compared to control (Figure 7). However, treatment with 10 and 20 ng/mL of vitamin D significantly increased 2-DG uptake by 17% (*p* < 0.01). Treatment with vitamin E alone significantly increased 2-DG uptake by 12% (*p* < 0.05). Additionally, the combination of both vitamins D (10 and 20 ng/mL) and E, increased the 2-DG uptake by 18% (*p* < 0.01).

### 3.6. ROS Assay

The production of ROS was significantly (*p* < 0.001) increased upon induction with 250 nM insulin (Figure 8). The presence of vitamin D, vitamin E and their combinations significantly reduced ROS production (*p* < 0.001) compared to the negative control. Upon treating the cells with 10 and 20 ng/mL vitamin D, the ROS production level was significantly reduced (*p* < 0.001) by 32.2 and 30.5%, respectively. Compared to the control, treatment with 200 ng/mL vitamin E alone showed a reduction of 37.6% (*p* < 0.01). Additionally, vitamins D (10 and 20 ng/mL) combined with vitamin E reduced the production of ROS by 46.5% (*p* < 0.001) and 27.4% (*p* < 0.01), respectively (Figure 8). 

### 3.7. Gene Expression of AD Markers

In order to determine the modulation of vitamins D and E on glycogen kinase activity and the pathogenesis of AD, the relative gene expression of *GSK3β* and *TAU* in the neuronal cells treated with various concentrations of vitamins D and E as well as the combination of both vitamins was determined. Treatment with vitamin D (10 ng/mL and 20 ng/mL) reduced the *GSK3β* expression level by 1.5- and 2-fold, respectively, compared to the negative control (Figure 9a). Treatment with vitamin E (200 ng/mL) and the combination of both vitamins also significantly reduced the expression level of *GSK3β* (*p* < 0.001) by almost 2-fold (Figure 9a). Treatment with vitamin D (10 and 20 ng/mL) and a combination of both vitamins significantly reduced the *TAU* expression level (*p* < 0.001) by almost 1-fold compared to the negative control (Figure 9b). Moreover, treatment with 200 ng/mL vitamin E significantly (*p* < 0.001) reduced the *TAU* expression level by 1-fold compared to the negative control (Figure 9b).

## 4. Discussions

The insulin-independent glucose uptake of the brain led to the belief that the brain is an insulin-insensitive organ that uses the hypothalamus as a critical regulator of insulin action [32]. Therefore, little attention has been given to brain-insulin signalling in the past. Previously, the hyperinsulin-induced decrease in cellular insulin activity has been reported in myocytes and adipocytes as well as in the nervous system, which in turn causes insulin resistance [28,33,34]. Today, it is clear that insulin is involved in many functions and signalling pathways in the brain. Recent studies indicate that AD and dementia are related to insulin impairment, which has a profound association with cognitive disorders. Insulin signalling dysregulation wreaks havoc primarily by impacting other mediator proteins [35]. To identify suitable agents that are useful in the alleviation of the condition of insulin resistance, the in vitro model of 250 nM insulin treatment for 16 h and re-challenged with 100 nM insulin in 30 min that mimics the state of insulin resistance and reflects the pathophysiological progression of the AD condition was adopted in this study. Our results showed that induction with hyperinsulin did not result in any harmful effect in the neuronal cells. Another study reported similar findings whereby induction with 1 μM insulin did not elicit any cytotoxic effects but promoted the growth of SH-SY-5Y neuronal cells [36]. Despite promoting growth, prolonged exposure to high insulin concentrations in this cell line has shown to induce insulin resistance [37]. More importantly, Pandini et al. [38] confirmed the presence of insulin markers in SK-N-SH neuronal cells and hence, attribute to the successful induction of insulin resistance model in this study. 

Insulin mediates its action through the insulin receptor substrate family. To develop a chronic insulin model, the concentration of 250 nM insulin was used to treat SK-N-SH cell lines for 16 h, and then the cells were re-challenged with insulin (100 nM) for 30 min as depicted in Figure 10. The compensation of chronic insulin causes the binding site of insulin receptors to become imbalanced, conferring the significant downregulation of the insulin signalling molecules *IR*, *PI3K*, *GLUT3* and *GLUT4*, as determined by the gene expression analysis (Figure 2a–d). Other studies indicate that prolonged exposure to high insulin concentrations increased IR degradation, which further dampened the insulin response [37]. and subsequently caused IR desensitization. The expression of phosphorylated *AKT* and *PI3K* are used as indicators of insulin activation in the *PI3K/AKT* pathway. *PI3K* plays a key role in insulin-stimulated glucose uptake in insulin-responsive tissues and was able to show whether chronic presence of insulin has any inhibitory effect after insulin challenge at different time points [28]. The level of *PI3K* expression in the current study was also decreased as a consequence of the reduction in *IR* expression after treatment with the high insulin concentration. Since *AKT* exerts its effect following *PI3K* activation, the decrease in *PI3K* expression leads to the reduction of *AKT* phosphorylation [39]. In the present study, induction with 100 nM insulin readily decreased *AKT* phosphorylation. Although 100 nM insulin was the lowest concentration adopted in this study, concentration as low as 5 nM have been observed to deplete insulin signalling in peripheral cells [40,41]. It has been shown that neuronal cell lines have a greater tolerance to the effects of hyperinsulin than do peripheral cells. Previous work on various neuronal cells also re-challenged cells with 100 nM insulin to determine the responsiveness of insulin resistance [28,42].

Decreased *AKT* phosphorylation inhibits the insulin-stimulated glucose transporter *GLUT4*. Although *GLUT4* is a glucose transporter in peripheral tissues, its expression has also been reported in the brain [30]. High insulin levels promote a reduction in the activity of the insulin signalling cascade and consequently reduces *GLUT4* translocation and/or expression [43]. Similar effects were also observed in SK-N-SH cells; the *GLUT4* expression levels were decreased after 16 and 24 h of insulin treatment (Figure 2d), resulting from the reduction in AKT phosphorylation. A reduction of *GLUT4* expression is related to the development of insulin resistance, since transgenic mice lacking *GLUT4* were shown to have insulin insensitivity [44]. *GLUT3* is considered the main glucose transporter in the brain. Although some studies demonstrated that *GLUT3* translocation is not sensitive to insulin stimulation, there is a study demonstrating that insulin regulates neuronal glucose uptake by promoting the translocation of *GLUT3* to the plasma membrane, suggesting that *GLUT3* might be involved in the insulin signalling cascade via the activation of *PI3K/AKT* pathway [45]. Although *GLUT3* has been reported to be insulin-independent, this mechanism only occurs in the brain as an endogenous protective mechanism against hypoglycaemia [46]. In parallel with our results, the *GLUT3* expression level in SK-N-SH neuronal cells was decreased (Figure 2c) as a consequence of the decrease in IR expression that led to the attenuation of the *PI3K/AKT* signalling pathway.

In this study, 10 and 20 ng/mL of vitamin D were able to upregulate IR, and hence, preserve insulin sensitivity similar to the negative control. Similarly, vitamin D is able to restore the function of *PI3K*, *GLUT3,* and *GLUT4* up to 50%. As suggested by previous studies, vitamin D upregulates *GLUT3* in the brainstem [47], where high glucose induces *GLUT4* translocation in 3T3L1 adipocytes [31]. Our findings indicate that vitamin D exhibits a high potency in the regulation of glucose homeostasis by improving the downstream key insulin signalling molecules and reversing hyperinsulinaemia [48]. This treatment effect occurs because the promoter region of the insulin receptor gene that regulates insulin receptor expression is adjacent to the vitamin D response element (VDRE), which is activated upon binding with the vitamin D receptor (VDR)-retinoic acid X-receptor (RXR) complex [49,50]. Besides, vitamin D has also been shown to regulate AMP-activated protein kinase (AMPK)-calmodulin, *AKT* or Notch signalling, which is responsible for the indirect effects in reducing intracellular oxidative stress. This explanation highlights the role of vitamin D by activating Ca^2+^/AMPK pathway, leading to glucose-sensing outcomes [51]. Thus, vitamin D has a rather direct effect on the insulin signalling cascade pathway. Additionally, vitamin D is also associated with the decreased production of advanced glycated end product (RAGE) receptors in hepatocytes, which is an oxidative stress marker that is closely associated with the pathogenesis of diabetes [52]. A study using hippocampal neurons HN9.10e demonstrates that vitamin D is responsible for the differentiation of hippocampal neurons that is regulated by VDR in nuclear microdomain [53]. In relation to previous findings, the presence of vitamin D receptors (VDRs) at hippocampus improved the glucose tolerance in high fat diet-induced rats [54]. It shows that VDRs in the neuronal cells could stimulate the *PI3K/AKT* signalling pathway to improve insulin sensitivity towards the cells in the presence of vitamin D, thus, modulates glucose homeostasis. Collectively, VDRs and vitamin D binding protein (DBP) have been detected in cerebrospinal fluid [55], macrophages, neurons and microglial in various *in vivo* and *in vitro* models [56].

In contrast, vitamin E did not show any significant improvement in restoring the sensitivity of any of the insulin signalling markers. However, the combination of vitamins D and E showed similar *GLUT4* expression levels compared to the group treated with vitamin D alone. This observation suggests the effectiveness of vitamin D in improving the immediate insulin signalling of neuronal cells. Thus, vitamin D could possibly act as a potential nutritional therapy that might be involved in stimulating *PI3K-AKT* signalling in insulin resistance, sensitizing the neuronal cells to downregulate the AD-like markers, particularly *GSK3β* and *TAU* gene expression.

The hypothetical evidence of insulin signalling dysregulation affecting hyper-phosphorylated AD markers by treating the hyperinsulin SK-N-SH cells with vitamins D and E was also validated. The chronic insulin resistance model shows significant suppression of *AKT* phosphorylation, suggesting the impairment of proteins in insulin signalling cascade pathways [57]. Clearly, the effect of vitamins D and E on increasing *AKT* phosphorylation has a high potency to restore insulin sensitization similar to the negative control (Figure 4). Previous research demonstrated that vitamin D increased the *AKT* phosphorylation cascade in apoptotic osteoblasts, which reduced caspase-3, an inflammatory cytokine [58]. A previous report also demonstrated that vitamin D suppressed cyclooxygenase-2 expression in response to inflammation in macrophages via the activation of *AKT* phosphorylation [59].

The binding of insulin to its receptor activates a phosphorylation cascade leading to the inhibition of *GSK3β*. When the activity of insulin signalling is disturbed, hyperphosphorylation of *GSK3β* occurs, which later causes an increase in *TAU* expression, a marker of AD [60]. In the present study, the decrease in insulin signalling markers was correlated with an increase in *GSK3β* and *TAU* expression levels. Not only does inhibiting insulin signalling cause an increase in *GSK3β* activation, but oxidative stress can also induce an increase in *GSK3β* expression [61,62]. Consistent with the above findings, *GSK3β* and *TAU* expression levels increased along with an increase in ROS production. The over-activation of *GSK3β* causes beta amyloid plaque aggregation, enhancing apoptosis and leading to neuronal loss in AD [61]. Interestingly, the increase in *GSK3β* and *TAU* expression upon insulin induction was reduced to normal levels with the treatment of both vitamins D and E. The antioxidant properties exhibited by vitamins D and E can potentially explain the reduction in both markers.

Vitamin E, in the form of tocopherol and tocotrienol, acts as an exogenous antioxidant to sustain the redox balance by the compensation of endogenous antioxidants inside the body [63]. Our results have clearly demonstrated that vitamin E in the form of TRF significantly downregulates ROS production after the induction of chronic insulin resistance, even though it did not exert any significant effect on the insulin signalling molecules. Tocotrienols exhibit good neuroprotective, radio-protective, anticancer, anti-inflammatory and lipid-lowering properties [23,64]. Chronic insulin resistance increases ROS production, leading to oxidative stress, mitochondrial dysfunction and activation of inflammatory cytokines. Oxidative stress inhibits the phosphorylation of insulin receptor substrate, leading to the inhibition of PI3K cascades signaling [65]. Similarly, vitamin E-treated L6 myotubes induced with insulin-stimulated hydrogen peroxide exhibited reduced ROS production [66]. Mechanistically, vitamin E acts as a protective agent to quench free radicals in cells leading to normalization of glucose uptake function of cells, preventing chain-breaking antioxidant properties. This reduces the activity of the endogenous antioxidant enzymes, such as catalase and superoxide dismutase, which plays more important roles in the endogenous antioxidant defense of human body [67]. 

However, the combination of both vitamins did not show a greater improvement in the insulin signalling cascade than treatment with vitamin D alone. It has been reported that the presence of more than one fat-soluble vitamin (at large doses) at one time may reduce the uptake of other fat-soluble vitamins by approximately 10 to 50% due to competition in the absorption across the intestinal endothelial cells [24]. Thus, to maximize the potency of vitamins D and E in improving the insulin signalling cascade and reducing ROS production, respectively, both vitamins can be given separately with a time gap at different concentrations. 

The results from this study present a unique opportunity for the prevention of ‘type 3 diabetes’, which is insulin resistance in the brain and the nervous system [6,68]. The results suggest that this validated *in vitro* model of hyperinsulinaemia in neuronal cells could be a simple, quick and physiologically-relevant model to study the complications of ‘type 3 diabetes’. However, *in vitro* studies are inadequate to ascertain in vivo mechanisms. Hence, in vivo studies should be conducted in the future. In light of the growing reports of the pervasiveness of insulin abnormalities in AD, there is an urgent need to search for therapeutic agents in AD. Thus, both vitamins D and E could be possible therapeutic agents that show great efficacy in improving neuronal insulin resistance and AD.

## 5. Conclusions

Treatment with vitamin D might improve the insulin signalling cascade by increasing all insulin signalling markers, which, in turn, increased glucose uptake that could reduce the risk of type 3 diabetes. However, vitamin E and the combination of vitamins D and E were not as effective as in improving insulin signalling compared to vitamin D alone. In addition, treatment with vitamins D and E reduced the production of ROS, where the expression of *GSK3β* was reduced to normal levels, leading to the normal expression of *TAU*. Collectively, vitamin E alone reduces oxidative stress that occurs during insulin resistance, as well as AD markers of *GSK3β* and *TAU*. These findings suggest a therapeutic role of vitamins D and E in managing the neurological disorders associated with diabetes since vitamin D increased the activity of IR, improving the insulin signalling cascade.

## Figures and Tables

**Figure 1 nutrients-11-02525-f001:**
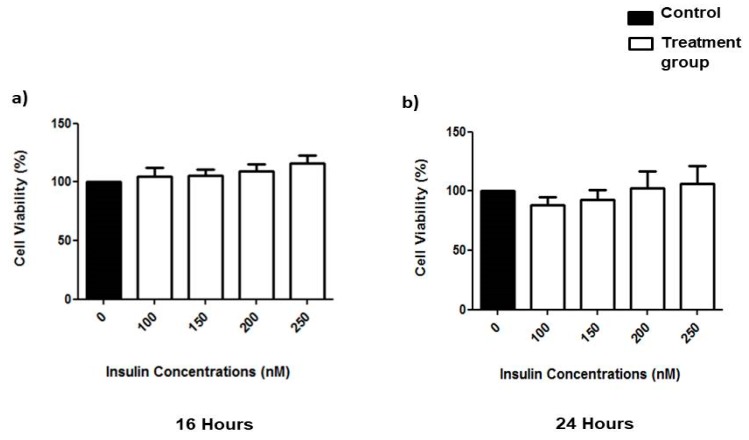
Effects of prolonged insulin induction on cell viability with different concentrations (100–250 nM) in SK-N-SH cell line incubated for (**a**) 16 and (**b**) 24 h, respectively. Cell viability was measured with an MTT assay. Data were expressed as mean ± SEM of three independent experiments.

**Figure 2 nutrients-11-02525-f002:**
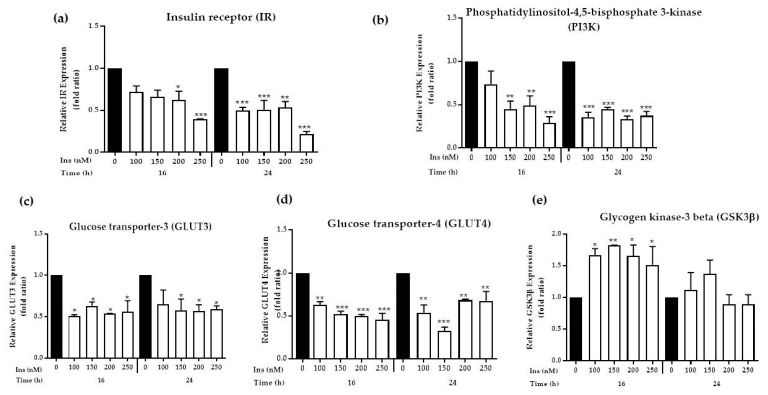
Effects of prolonged insulin induction on insulin signal transduction cascade in SK-N-SH cells line. Serum deprived cells were incubated in the absence (control) or presence of insulin for 16 and 24 h, and challenged with 100 nM insulin for 30 minutes. Total RNAs were prepared and subjected to reverse transcriptase. The resulting cDNAs were used to perform Real-time PCR as described in Section 2.5 using specific primers for (**a**) *IR*; (**b**) *PI3K*; (**c**) *GLUT3*; (**d**) *GLUT4* and (**e**) *GSK3β*. *GAPDH* was used to normalize the expression level. Data were expressed as mean ± SEM of three independent experiments. Significant values were expressed as * *p* < 0.05, ** *p* < 0.01 and *** *p* < 0.001 compared to the negative control group.

**Figure 3 nutrients-11-02525-f003:**
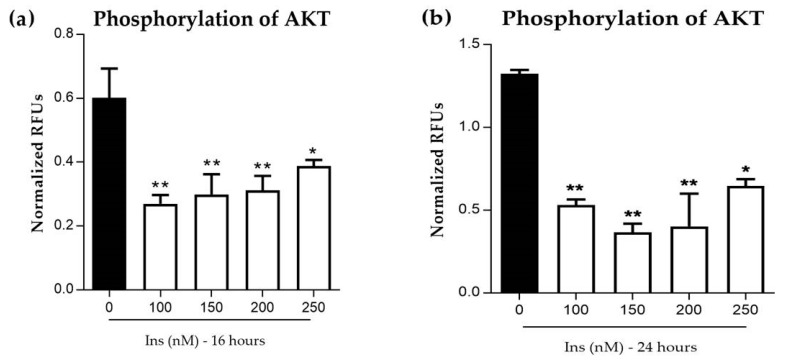
Effects of prolonged insulin induction with 100–250 nM concentrations on protein kinase B *(AKT)* phosphorylation. (**a**) A relative (p)-*AKT* phosphorylation was measured using Phospho-*AKT* Immunoassay ELISA at 16 h of exposure with insulin; while (**b**) at 24 h and both were re-challenged with 30 min induction. Relative p-*AKT* levels were normalized to total *AKT* and are shown as fold increase over control (0 nM of insulin). All data were expressed as mean ± SEM of three independent experiments. Significant values were expressed as * *p* < 0.05 and ** *p* < 0.01 compared to the negative control group.

**Figure 4 nutrients-11-02525-f004:**
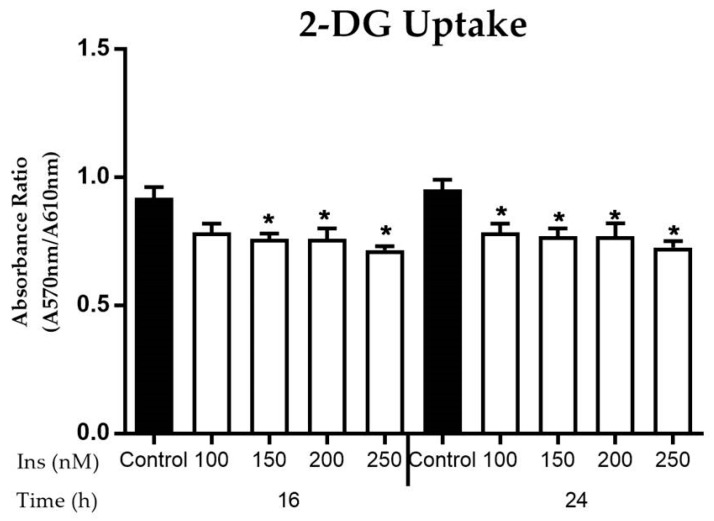
Effects of insulin exposure at 150–250 nM concentrations on 2-D-Glucose (2-DG) uptake after incubation at 16 and 24 h. The cells were subjected to prolonged exposure at 16 and 24 h with re-challenged for 30 min. The induced groups were compared with the control group (0 nM of insulin). Results were expressed as mean ± SEM of three independent experiments. Significant values were expressed as * *p* < 0.05 compared to the negative control group.

**Figure 5 nutrients-11-02525-f005:**
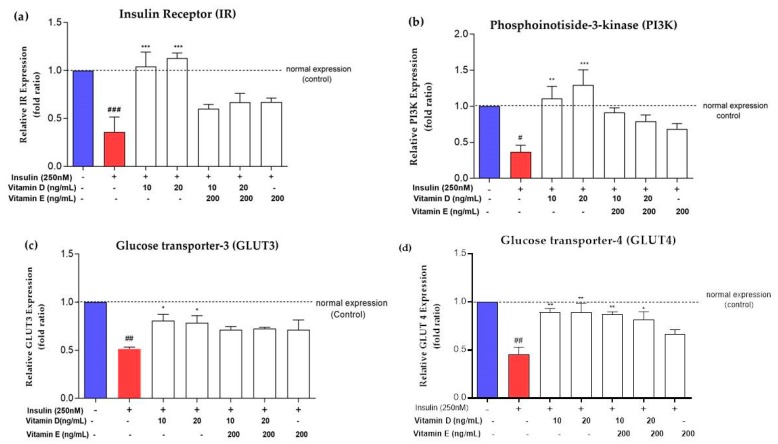
Gene expression levels of (**a**) *IR* (**b**) *PI3K* (**c**) *GLUT3* and (**d**) *GLUT4* after treatment with vitamins D and E at different concentrations. Red bar represents the expression level upon induction with 250 nM insulin for 16 h. White bar represents the expression level treated with vitamins D and E for 24 h after induced with 250 nM insulin. Relative expression of *IR*, *PI3K*, *GLUT3,* and *GLUT4* were normalized with *GAPDH*. Data were expressed as mean ± SEM of three independent experiments. Significant values were expressed as * *p* < 0.05, ** *p* < 0.01, *** *p* < 0.001 compared to the negative control group and # *p* < 0.05, ## *p* < 0.01, ### *p* < 0.001 compared to the control group.

**Figure 6 nutrients-11-02525-f006:**
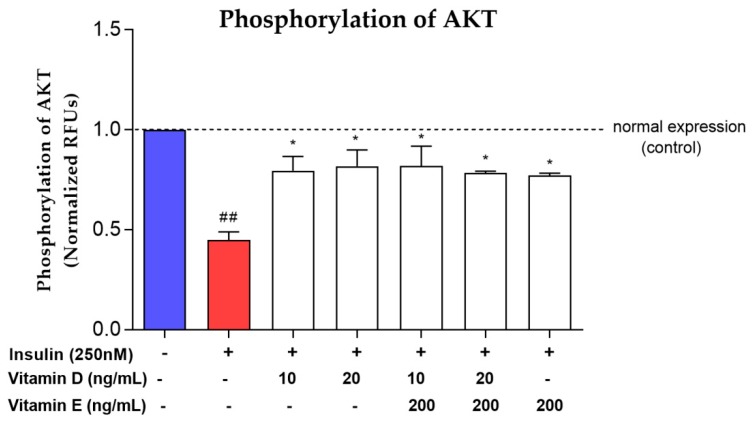
*AKT* phosphorylation level in insulin-resistant cells upon treatment with various concentrations of vitamins D and E. Red bar represents *AKT* phosphorylation level upon induction with 250 nM insulin for 16 h. White bar represents *AKT* phosphorylation level upon treated with vitamins D and E for 24 h after induced with 250 nM insulin. Relative p-*AKT* phosphorylation levels were normalized to total *AKT* and were expressed as fold increase over control. Data were expressed as mean ± SEM of three independent experiments. Significant values were expressed as * *p* < 0.05 compared to the negative control group and ## *p* < 0.01 compared to the control group.

**Figure 7 nutrients-11-02525-f007:**
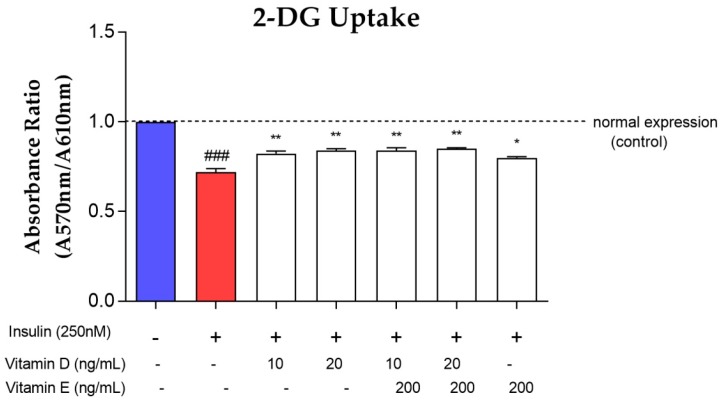
2-DG uptake level in insulin-resistant cells upon treatment with various concentrations of vitamins D and E. Red bar represents the 2-DG uptake level upon induction with 250 nM insulin for 16 h. White bar represents the 2-DG uptake level upon treatment with vitamins D and E for 24 h after induced with 250 nM insulin. Data were expressed as mean ± SEM of three independent experiments. Significant values were expressed as * *p* < 0.05, ** *p* < 0.01 compared to the negative control group and ### *p* < 0.001 compared to the control group.

**Figure 8 nutrients-11-02525-f008:**
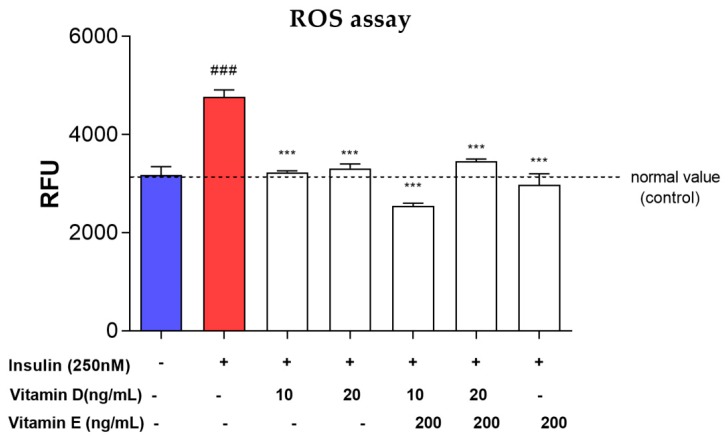
Reactive Oxygen Species (ROS) production level in insulin resistance model to determine the antioxidant potential. Red bar represents the ROS production level upon induction with 250 nM insulin for 16 h. White bar represents the ROS production level after treated with vitamin D and E for 24 h after induced with 250 nM insulin. Data were expressed as mean ± SEM of three independent experiments. Significant values were expressed as *** *p* < 0.001 compared to the negative control group and ### *p* < 0.001 compared to the control group.

**Figure 9 nutrients-11-02525-f009:**
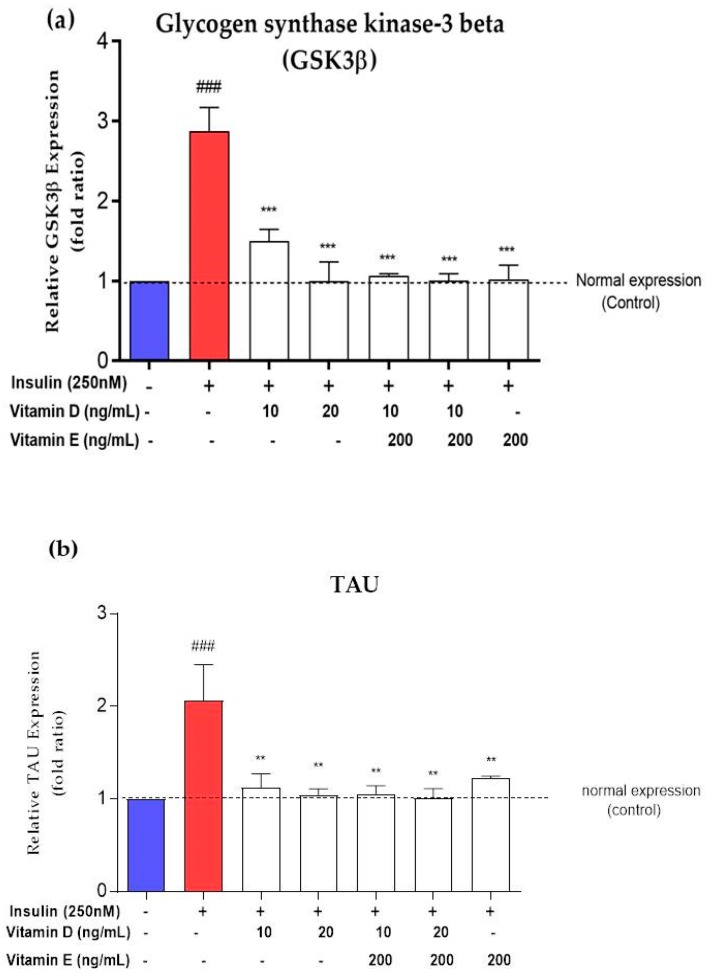
(**a**) *GSK3β* and (**b**) *TAU* expressions level in insulin-resistant cells upon treatment with various concentrations of vitamins D and E. Red bar represents expression level upon induction with 250 nM insulin for 16 h. White bar represents expression level treated with vitamins D and E for 24 h after induced with 250 nM insulin. Relative *GSK3β* and *TAU* expressions were normalized to *GAPDH*. Data were expressed as mean ± SEM of three independent experiments. Significant values were expressed as ** *p* < 0.01, *** *p* < 0.001 compared to the negative control group and ### *p* < 0.001 compared to the control group.

**Figure 10 nutrients-11-02525-f010:**
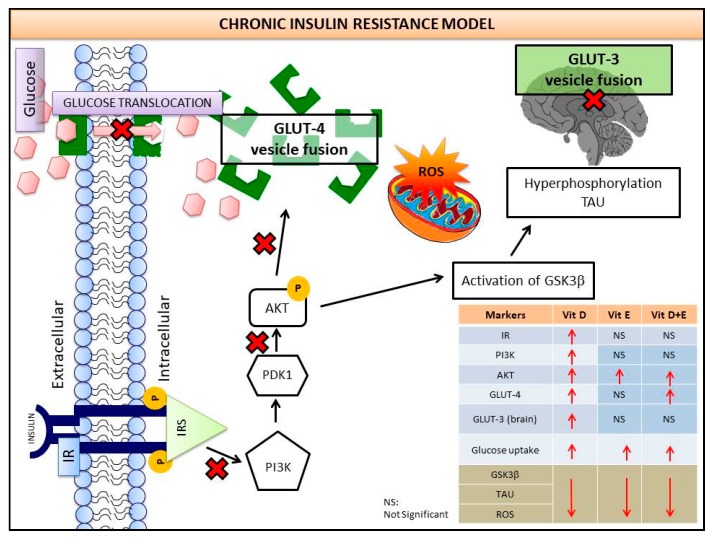
Continuous impaired insulin signalling causes the activation of AD markers (*GSK3β* and *TAU* genes), thus increasing the amyloid beta aggregations leading to AD. Current study reports the role of vitamin D, E and the combinations of vitamins D and E in restoring insulin signalling to normal physiological condition by downregulating *GSK3β* and *TAU* genes, eventually reducing the risk of AD.

**Table 1 nutrients-11-02525-t001:** Primers for the genes of interest.

Gene of Interest	Forward Primers	Reverse Primers
*IR*	CAACATTCGAGGAGGCAACAATC	CTCGAATCAGACGTAACTTCCGG
*PI3K*	TGGATGCTCTACAGGGCTTT	GTCTGGGTTCTCCCAATTCA
*GLUT3*	TCCCCTCCGCTGCTCACTATTT	ATCTCCATGACGCCGTCCTTTC
*GLUT4*	CCCTCAGAAGGTGATTGAACAG	AGAGATGATACCAATGAGGAAGG
*GSK3β*	CTAACACCACTGGAAGCTTGTGC	GATGGTAGCCAGAGGTGGATTAC
*TAU*	ACCTCCAAGTGTGGCTCATTAG	GGACGTGGGTGATATTGTCCAG
*GAPDH*	CAACTACATGGTTTACATGTTC	GCCAGTGGACTCCACGAC

*IR*, insulin receptor; *PI3K*, phosphatidylinositol-4,5-bisphosphate 3-kinase; *GLUT3*, glucose transporter 3; *GLUT4*, glucose transporter 4; *GSK3b*, glycogen synthase kinase-3 beta; *TAU*, tau; and *GAPDH*, glyceraldehyde-3-phosphate dehydrogenase.

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
