# Peer review of "Vitamins D and E Stimulate the PI3K-AKT Signalling Pathway in Insulin-Resistant SK-N-SH Neuronal Cells"

_nutrients, 2019, doi:10.3390/nu11102525_

Round 1

Reviewer 1 Report

The manuscript is well written. The authors show, using cell culture models of insulin resistance, that treatment with Vitamin D and E improve insulin sensitivity and expression of certain key genes. However, the whole paper is based on mRNA expression data that are not corroborated at the protein level. The function of many of the proteins listed in this study are modulated postranslationally. For example, Tau hyperphosphorylation triggers the pathogenesis in AD. Please show Western for protein data for GSK3beta and Tau as well as their phosphorylation status to support the link to AD pathophysiology. In figure 7, the legend says protein expression, however, there are no methods to quantify protein expression for GSK3beta and Tau (Western, anitbodies, etc). The quantification of these were for mRNA from qPCR as suggested in section 2.5. Please amend legend for Figure 7 and provide protein expression data. 

Please show Western data for AKT phosphorylation along with the quantification from the ELISA. Are there morphological changes to the cells with insulin induction since the induction of insulin insensitivity increased ROS levels in these cells? Considering these cells originated from a neuroblastoma and epithelial in nature, how does the data relate to neuronal cells and the pathogenegesis of AD?

Reviewer 2 Report

Zaulkfalli and colleagues investigated the effect of vitamin D and E, alone or in combination, in neuronal cells SK-N-SH after inducing an insulin resistance condition. To this aim, the authors performed several in vitro experiments, including the evaluation of gene related to insulin pathway, glucose uptake and Alzheimer’s disease (AD) by RT-PCR and ELISA. They concluded that vitamin D and E could be used as potential therapeutic agent to restore insulin sensitivity in neuronal cells, delaying the progression of AD.

The topic it interesting and the experiments are well performed and described. However, some comments need to be addressed to improve the manuscript.

The topic of the paper is Alzheimer’s disease. However, the authors used SK-N-H, a neuroblastoma cell line. Are those cells a good model also for AD? Why the authors chose this cell line instead of other cell lines more often used as model of AD? Please, explain this point in the text. There are several parts that repeating same notion in the entire text. Please, avoid and correct them in the entire manuscript. The bibliography is written in a wrong order in the entire text (e.g. page 3, line 101, reference number 56 and 60 appear soon after reference 21). Please, reorganize the bibliography according with the authors guidelines of the journal. Statistical analysis. It is complicate to follow the meaning of a, aa, b, bb, *, **, ect. In each figure the p value is expressed in a different way. Moreover, from the figures, the significant of a and b seems vice versa that those written in the text. It creates a lot of confusion. Please, if you want use only symbols, choose only one (i.e *, **, etc) and explain in the figure legend which group you compare. Other important point is that the results are expressed only with p value (and only in the figures and not in the text). However, it is well known that only o value is not so important. The authors should add the differences of mRNA concentrations in fold-change at least for the most significant results (as they did only in one part of the paper). The first part of result should be entirely revised (page 6-8). First of all, in this part you should write only the obtained results. The same is also true for the figure legends. Thus, delete any comments on pathways and results and write them in the discussion. Second, graphs are discussed in the text in a different order than the figures (i.e. figure 2a, 2b and then 2e, 2d, 2c). This creates a lot of confusion for the readers. Use the same order for graphs and text. Same for figure 2 and 3, where the graphs representing the mRNA expression levels are in a different order. Please, use the same sequence (e.g. figure a=IR, figure b=PI3K, etc.). Moreover, use a title for each graph indicating the name of the evaluated gene. This will be very helpful for the readers. Other comments: page 6, line 250: from the figure, PI3K is significantly expressed also from 150 nM insulin. Figure 2e, line 252: you should specify that the results were obtained only after 16h of treatment (as for the following figure). You should better discuss this discrepancy between 16 and 24 hours in the discussion part. Figure 2c, lines 255-256: do no comment here the results with hypothesis, just write what you observed. Results page 8: the authors should better underline that vitamin E correlates only with GLUT4, and does not influence the other investigated genes. Figure 3: Why do you not evaluated also GSK3beta as in the previous experiment? Discussion: this part is too long. Please, rewrite avoiding repetition. Moreover, the uthors cannot compare the results obtained in SK-N-H cells with those obtained in different cell lines, as N2A cells (Gupta A et al. Neuropharmacology 2011) and 3TL1 adipocytes (Sangeetha KN et al, Phytomed 2013). Thus, they cannot conclude that vitamin D and vitamin E are more effective than metformin and rosiglitazone for the treatment pf hyperglycemia. Similarly, they cannot affirm that metformin could be replace by these two vitamins: this is a very strong conclusion that has no evidence. On the contrary, the authors did not underline in the conclusion that vitamin E is not effective as vitamin D in the modulation of the evaluated genes, and that the combination of the two vitamins is not more effective than the vitamin D alone. Thus, the conclusion should be focus on this point and the take home message should be that “vitamin D alone might help for the treatment of hyperglycemia and might prevent the type 3 diabetes, whereas vitamin E is less effective and the combination of the two vitamin did not improve the effect of vitamin D alone”. Please, reformulate the discussion and the conclusion in this perspective. I would suggest English revision along the entire manuscript. Please, write the full name of all abbreviations the first time that you mention them (page 2, line 82). Page 2, lines 91-94. Did you purchase primers for insulin receptor, GLUT4, etc.? Please, specify. Page 4, line 147. Markers of what? Moreover, you should specify that you evaluated the expression of the described genes at RNA level. Page 4, line 175. Was insulin dissolved in DMSO or cell medium? In the first case, is better to indicate the negative control as DMSO instead of 0 nM insulin.

Round 2

Reviewer 1 Report

While there are still concerns on whether the changes in gene expression correlate with protein levels and function, the authors have addressed the concerns raised.

Reviewer 2 Report

The authors replied to all the questions, and improved the manuscript. In my opinion, the effects of vitamin E continues to be overestimate in the text. However, the manuscript can be published in this form.